# A multi-group path analysis of medication documentation quality using cross-sectional survey data: Impact of leadership, job satisfaction, patient-related burnout, and patient safety culture

**Nikoloz Gambashidze**[1]☯*, **Anke Wagner**[2]☯, **Tanja Manser**[1,3], **Monika A. Rieger**[2], **Peter Martus**[4], **Antje Hammer**[1], on behalf of the WorkSafeMed Consortium[¶]

1 Institute for Patient Safety, University Hospital of Bonn, Bonn, Germany, 2 Institute of Occupational and Social Medicine and Health Services Research, University Hospital of Tübingen, Tübingen, Germany, 3 FHNW School of Applied Psychology, University of Applied Sciences and Arts Northwestern Switzerland, Olten, Switzerland, 4 Institute for Clinical Epidemiology and Applied Biometry, University Hospital of Tübingen, Tübingen, Germany

¶ Membership of the WorkSafeMed Consortium is provided in the Acknowledgments.
☯ These authors contributed equally to this work.
* Nikoloz.Gambashidze@ukbonn.de

## Abstract

Medication-related adverse events are a major problem for patient safety and quality of care. This study explored the effects of leadership, job satisfaction, patient-related burnout and patient safety culture, measured through surveys of frontline workers, on documentation quality. Perceptions of physicians and nurses were surveyed using established instruments including the Transformational Leadership Inventory, the Copenhagen Psychosocial Questionnaire, the Copenhagen Burnout Inventory, and the Hospital Survey on Patient Safety Culture. Documentation quality was evaluated using retrospective review of patient records with the MediDoq instrument. Data from 24 departments, covering 282 physicians, 417 nurses, and 802 patient records, were analysed through a multi-group (physicians and nurses) path analysis to evaluate the theoretical model. The theoretical model revealed satisfactory fit, explaining about 76–80% of the variance in Documentation Quality for physicians and nurses. Perceived Patient Safety had a significant effect on Documentation Quality only for nurses. The analyses revealed differences between professional groups. Based on these results, (i) medication documentation quality may be affected by various organizational factors; (ii) perceived Patient Safety reported by the nurses may mediate some of these effects; (iii) the effects of various organizational factors on documentation quality may vary significantly across professional groups.

**Data availability statement:** Due to the sensitive nature of the data collected in the WorkSafeMed study, involving personal information strictly governed by national and regional data protection laws, the dataset cannot be made available in the public domain. Data will be stored in accordance with national and regional data security standards. Access to the data may be requested from the Institute for Patient Safety at the University Hospital Bonn (ifps@ukbonn.de). All requests will be subject to ethical review and compliance with institutional and legal data protection standards.

**Funding:** The „WorkSafeMed" study was funded by the Federal Ministry of Education and Research (FKZ 01GY1325A, awarded to MAR, and 01GY1325B, awarded to TM). In addition, the work of the Institute of Occupational and Social Medicine and Health Services Research Tuebingen is supported by an unrestricted grant of the Employers´ Association of the Metal and Electric Industry Baden-Wuerttemberg (Suedwestmetall). We acknowledge additional financial support by the German Research Foundation and the Open Access Service Center of the University of Bonn, as well as the administrative support by the DLR Project Management Agency. The funders had no role in study design, data collection and analysis, decision to publish, or preparation of the manuscript.

**Competing interests:** The authors have declared that no competing interests exist.

## Introduction

Medication treatment is the most commonly used therapy, a daily routine in a hospital setting, involving both nurses and physicians. Unfortunately, medication-related adverse events (MRAEs) are still among the most common healthcare related adverse events. A meta-analysis found that for hospital settings, as high as 19% of all hospitalized patients may be affected by MRAEs [1]. A study involving junior doctors revealed 7.5% of prescriptions with possible errors, involving up to 36% of patients [2], which is in line with another study reporting 8.8% of error rate in all prescriptions [3]. More than 80% of identified medication errors may reach patients, though most caused no harm [4]. The review by Schwendimann et al. found medication errors among the three most frequent adverse events in hospital settings [5].

The medication process comprises several error-prone steps, which require clear and transparent documentation in patient records to avoid MRAEs. Incomplete or incorrect medication documentation hinders the information flow and may contribute to various errors, affecting the care process and endangering patients [6]. Previous analyses identified considerable lack in medication documentation quality [7]. This was particularly the case when documentation was postponed (e.g., delays due to interruptions, or postponed documentation of verbal prescriptions) or additional efforts were required for documentation (e.g., documentation at the end of shifts because patient records were not accessible at the time of the administration of the medication). Proper, timely, and clear documentation can reduce MRAEs and improve patient safety [4,8,9].

In order to develop strategies to improve documentation quality, it is crucial to identify and study the factors affecting it. Different organization- and individual-related factors may influence documentation quality [2]. In this study, we focus on the organizational factors, including leadership, job satisfaction, patient-related burnout, and aspects of patient safety culture, as potential predictors of the medication documentation quality.

Leadership, whether it's hospital management or direct supervisors within hospital units, affects organizational outcomes by setting directions, defining priorities, allocating resources, and establishing the required policies and procedures for implementing improvement strategies. Leaders create a context, which encourages and supports frontline professionals to improve patient safety [10–12]. Moreover, previous studies have identified high workloads, stress and mental health issues having a negative impact on patient safety [13,14]. In addition, burnout has been associated with lower perception of patient safety among employees [15].

Patient safety culture is a relatively stable, multidimensional construct based on employees´ shared values and norms regarding patient safety [16], which is considered to facilitate improvements in patient safety [17,18]. A literature review on patient safety culture in relation to patient safety and quality of care-related outcomes found that frontline professionals' perceptions of patient safety were one of the two most frequently used outcome measures to evaluate patient safety [10]. Indeed, physicians' and nurses' perceptions and evaluation of patient safety are considered valid

proxies for patient safety, which is found to be positively associated with a number of organizational factors, including leadership [19], working conditions [19,20], and various aspects of patient safety culture [19,21]. The second frequently used outcome measure was medications errors. Documentation quality is directly related to these outcome measures and therefore the focus of this study.

The model, developed based on previous analysis [22] and theoretical considerations, is presented in Fig 1. On the left are various organizational factors representing leadership, job satisfaction, and burnout, as well as different aspects of patient safety culture, all measured using surveys of frontline staff. On the right are the two outcome measures: Documentation Quality and perceived patient safety. We evaluated the potential of perceived patient safety to act as a mediator, having a direct effect on Documentation Quality. Moreover, because studies involving frontline hospital staff consistently show significant differences between the perceptions of physicians and nurses from the same departments [23–25], we conducted the analysis for these two professional groups separately.

Currently, there is a limited evidence base examining the factors affecting the medication documentation quality. This study seeks to address this gap by exploring how leadership, job satisfaction, patient-related burnout, and patient safety culture contribute to documentation quality within hospital settings. More specifically, we aimed to answer the following research questions:

Q1: To what extent can leadership, job satisfaction, patient-related burnout and patient safety culture predict medication documentation quality?

Q2: To what extent does perceived patient safety mediate this relationship?

Q3: Are there any differences in this regard between physicians and nurses?

**Fig 1. Conceptual framework for the path model.**

## Methods

### Study design

The data for the following analyses come from the cross-sectional, bi-centric, mixed-methods project "Working Conditions, Safety Culture and Patient Safety in Hospitals – What Predicts the Safety of the Medication Process" (Work-SafeMed) [22]. The project aimed to examine the influence of working conditions and safety culture on patient safety, with a focus on medication process in hospitals. Data collection included staff surveys and retrospective chart reviews in two German university hospitals. We included all inpatient units treating at least 500 patients per year to ensure sufficient analysable patient records for the retrospective chart review. Intensive care and psychiatric units were excluded due to their distinct documentation practices, which are not directly comparable to general inpatient units. This resulted in a total of 37 eligible departments. We invited all physicians and nurses from the eligible departments to participate in the survey. An a priori sample size calculation was not conducted. We used a standardized paper-based survey to measure staffs' perceptions on leadership, job satisfaction, patient-related burnout and patient safety culture. Survey data was collected from April to July 2015. We did send at least one oral or written reminder one month after surveys were distributed. To measure various aspects of medication documentation quality, we conducted a retrospective chart review of patients treated from April to July 2015. This review was carried out between April and September 2016, during which reviewers had full access to patient records. After the data collection, all analyses were conducted using anonymized data.

### Measures and data collection

**The medication documentation quality.** To measure medication documentation quality, the outcome variables in our model, we used the MediDocQ instrument [7]. This is an instrument for retrospective chart review that captures data on: (1) completeness of information concerning prescription, administration, and adjustments of medications, including documentation of verbally communicated and pro re nata (PRN) medications; (2) quality of transcriptions (e.g., from prescription chart to medication administration chart); and (3) compliance with chart structure, legibility, handling of deletions and corrections. The instrument consists of 54 appraisal criteria comprising 11 generic and five additional aspects (indices) associated with documentation quality. Generic indices apply to all records (e.g., presence of date, time and signature), while additional aspects are relevant only to some cases (e.g., PRN medications). Values of indices range from 0 to 1, with values above 0.75 indicating higher medication documentation quality. The instrument is described in detail elsewhere [7]. In this analysis, we used six generic criteria applicable to most patient records. Specifically, we selected three criteria associated with medical prescriptions: "O1: Date, time, signature", "O2: Prescription Completeness," and "O3: Legibility of prescription chart"; and three criteria associated with the medication administration: "O4: First medication administration chart", "O5: Transcription of adjustments" and "O6: Legibility of administration chart". The first three criteria, referring to the quality of medication prescription chart, are mainly in the domain of the physicians' responsibility; while the later three, referring to medication administration charts, reflect responsibilities of nurses. Before the analyses, one item from "O3: Legibility of prescription chart" and another from "O6: Legibility of administration chart" were removed ("Pencil documentation in prescription chart" and "Pencil documentation in medication administration chart") due to a high number of missing answers.

**Surveys of frontline stuff.** The predictor variables in the model were captured with items from validated and well-established instruments. Specifically, we used the short scale of Transformational Leadership (TLI-short) [11,26] to measure perceptions of leadership; the German version of the Copenhagen Psychosocial Questionnaire (COPSOQ) [27] to measure job satisfaction; the Copenhagen Burnout Inventory (CBI) [28] to measure patient-related burnout; and the German version of Hospital Survey on Patient Safety Culture (HSPSC-D) [21] to measure various aspects of patient

safety culture. Two scales form HSPSC-D were duplicated to evaluate the behaviour and support of direct supervisors and the hospital management separately.

As a mediating variable in the model, we choose "Perceived Patient Safety", a factor combining two scales and two single items taken from the HSPSC-D. We calculated mean of "Overall Perceptions of Patient Safety" (outcome dimension, HSPOC-D), "Frequency of errors reported" (outcome dimension, HSPOC-D), "Overall Patient Safety Grade" (single item outcome, HSPOC-D) and "Overall Medication Safety Grade" (single item outcome, modified from HSPOC-D). The selection of these items was based on previous analyses in the project, where this factor proved to be a reliable outcome measure [29]. For the analysis of HSPSC results, we used mean scores and standard deviations instead of positive response rates. This approach was chosen to ensure consistency with other survey-based variables in the path model, facilitating direct comparisons across constructs.

## Statistical analyses

Prior to data analyses, respondents with missing values >30% were excluded due to limited data quality. Remaining missing values in the survey data were imputed with NORM 2.03 software using the Expectation-Maximization-algorithm [30]. Descriptive statistics were calculated to provide frequencies, means, and standard deviations (SDs) for both survey and chart review data. To facilitate the analysis of data from two distinct sources, individual employee surveys and patient-level record reviews, we aggregated the survey data at the departmental level. This approach enabled us to match these datasets and conduct joint analyses. Survey data were aggregated at the department level separately for nurses and for physicians. Next, we merged aggregated survey data with the chart review data at the department level. This resulted in two data sets – one matching the chart review data with perceptions of nurses, and another with perceptions of physicians. The analyses included only complete chart review data from departments where both physicians and nurses took part in the survey. We used multi-group (physicians and nurses) path analysis to evaluate the theoretical model presented in Fig 1 (path diagram available in the Supplementary Materials, S1 Fig in S1 Appendix). In the model, organizational factors such as transformational leadership, job satisfaction, patient-related burnout, and various aspects of patient safety culture were included as predictors of medication documentation quality. Perceived patient safety was specified as a potential mediator of the relationship between organizational predictors and documentation quality. Separate models were estimated for nurses and physicians using multi-group path analysis to explore profession-specific relationships. Absolute and incremental fit indices were calculated to evaluate model fit [31,32]. We used $R^2$ as indication of the percent of variance explained by the model. To evaluate mediation, we calculated standardized direct and indirect effects of predictors on outcome variables and used bootstrapping to estimate two-tailed significance of these effects. These effects were calculated for the two professional groups separately. Data were analysed using IBM Statistics SPSS (Version 25) and AMOS (Version 25) for Windows.

## Ethics and confidentiality issues

Ethics approval was obtained from the ethics committees at the two participating university hospitals (Reference numbers #350/14 and #547/2014BO1). Each partner complied with confidentiality requirements according to German law. At one hospital, the study was classified as a quality improvement initiative, and therefore, informed consent from patients was not required. At the second hospital, all patients provided written informed consent. Prior to completing the questionnaire, study participants received detailed information about the study, highlighting that participation was voluntary and anonymous. Participants were asked to proceed with the questionnaire only if they consented to participate. All data were analysed anonymously.

## Patient and public involvement

Patients and the general public were not involved in the design or implementation of this study.

**Table 1. Descriptive characteristics of survey data.**

| Survey data aggregated for (n = 24) departments | Scale Range[a] | Physicians Mean (SD) | Nurses Mean (SD) | p-value[b] |
|---|---|---|---|---|
| **PREDICTORS** | | | | |
| **P01: Transformational leadership** | 1.00-5.00 | 3.15 (0.38) | 3.16 (0.34) | 0.91 |
| **P02: Job satisfaction** | 0.00-100.00 | 72.37 (5.40) | 67.21 (4.41) | <0.01* |
| **P03: Patient-related burnout**[c] | 0.00-100.00 | 27.98 (5.76) | 35.93 (7.21) | <0.01* |
| **Patient Safety Culture** | | | | |
| P04: Individual influence on patient safety[d] | 1.00-5.00 | 3.11 (0.35) | 2.87 (0.32) | 0.02* |
| P05: Staffing | 1.00-5.00 | 2.77 (0.38) | 2.41 (0.38) | <0.01* |
| P06: Organizational learning | 1.00-5.00 | 3.11 (0.32) | 3.03 (0.25) | 0.33 |
| P07: Feedback and communication about error | 1.00-5.00 | 3.31 (0.46) | 3.43 (0.35) | 0.30 |
| P08: Handoffs and transitions | 1.00-5.00 | 2.91 (0.34) | 3.13 (0.26) | 0.02* |
| P09: Hospital management support for patient safety | 1.00-5.00 | 2.94 (0.42) | 2.69 (0.32) | 0.02* |
| P10: Hospital management behaviours related to patient safety | 1.00-5.00 | 3.21 (0.34) | 3.21 (0.28) | 0.94 |
| P11: Direct supervisor support for patient safety | 1.00-5.00 | 3.51 (0.36) | 3.52 (0.31) | 0.88 |
| P12: Direct supervisor behaviours related to patient safety | 1.00-5.00 | 3.04 (0.42) | 2.89 (0.30) | 0.14 |
| **MEDIATOR** | | | | |
| M01: Perceived patient safety | 1.00-5.00 | 3.19 (0.31) | 3.03 (0.27) | 0.06 |

Note: [a] – All survey scales are coded in positive direction, so that higher scores correspond to more positive evaluation, except for Patient-related burnout ([c]), where high scores correspond to more burnout, and thus to more negative outcomes. [b] – p-value based on independent samples t-test, comparing department level data of physicians and nurses.

[d] – Single item. * – p < 0.05.

## Results

### Descriptive results

We received 995 out of 2512 distributed surveys (response rate = 39.6%). The sample included 57.0% nurses and 38.3% physicians. The mean age of the participants was 37.67 years (SD = 10.69), and their average professional experience was 13.49 years (SD = 10.91). These data were aggregated for the 37 departments separately for nurses and for physicians. The number of participating nurses per department varied between 0 and 50, with a median of 14. The number of physicians per department varied between 0 and 23, with a median of 10.

1361 patient records from 29 departments were reviewed using MediDocQ instrument. Of these, 1291 records involved at least one documented medication. In the final combined dataset, we included 802 patient records with complete data on six outcome variables (O1-O6) from 24 departments where the survey data was available for both physicians and nurses. The analysis incorporated survey data of 282 physicians and 417 nurses from these 24 departments. Physicians and nurses in 24 departments had somewhat similar perceptions of Transformational leadership (P01) (physicians mean = 3.15, SD = 0.38; nurses mean = 3.16, SD = 0.34). On average, nurses reported lower Job satisfaction (P02) and higher Patient-related burnout (P03) compared to physicians. The Patient safety culture indices for both groups varied between 2.41 and 3.52 (on a scale of 1–5), with both physicians and nurses scoring Staffing (P05) the lowest, and Direct supervisor support for patient safety (P11) the highest. Descriptive results on survey data used in the path model as independent variables are provided in Table 1.

The Documentation Quality measured by chart review varied between patients. Five out of six evaluated quality aspects reached an acceptable level of >0.75 (on a scale of 0–1). Documentation of date, time, signature (O1) had the lowest mean score of 0.67 (SD = 0.24). Descriptive characteristics of the reviewed patient records are presented in Table 2.

**Table 2. Descriptive characteristics of reviewed patient records.**

| Chart review data based on review of (n = 802) patient records | Scale Range[a] | Mean (SD) |
|---|---|---|
| **OUTCOME** | | |
| **Medication prescription** | | |
| O1. Date, Time, Signature | 0.00-1.00 | 0.67 (0.24) |
| O2. Prescription Completeness | 0.00-1.00 | 0.80 (0.15) |
| O3. Legibility of prescription chart | 0.00-1.00 | 0.87 (0.16) |
| **Medication administration** | | |
| O4. First medication administration chart | 0.00-1.00 | 0.85 (0.12) |
| O5. Transcription of adjustments | 0.00-1.00 | 0.77 (0.35) |
| O6. Legibility of administration chart | 0.00-1.00 | 0.89 (0.16) |

Note: a – The scores of chart review data represent the quality of various aspects of medication documentation, measured from 0 to 1 (from low to high quality).

## Results on the path model

The path analysis, which included 802 patient records from 24 departments and aggregated survey data for physicians and nurses separately (two professional groups), resulted in a reasonable fit with the data. Most fit indices met the desired benchmarks: $Chi^2 = 806.7$, degrees of freedom (df) =148, $p < 0.001$, $Chi^2/df = 5.45$, The root mean square error of approximation (RMSEA)=0.053, the goodness of fit index (GFI)=0.951, the adjusted goodness of fit index (AGFI)=0.873, the normed fit index (NFI)=0.972, the non-normed fit index (NNFI)=0.947, and the comparative fit index (CFI)=0.977 (Supplementary Materials, S2 Table in S2 Appendix).

Table 3 presents the direct and indirect effects of predictors and the mediator on outcome variables. The size and significance of the indirect effects measured in the analysis depended on the size and significance of the direct effect of the mediator Perceived Patient Safety on Documentation Quality. In case of nurses, Perceived Patient Safety (M01) had a

**Table 3. Standardized effects on Documentation Quality evaluated in path model separately for nurses and physicians.**

| | Physicians | | | Nurses | | |
|---|---|---|---|---|---|---|
| | Direct Effect | Indirect Effect | Total Effect | Direct Effect | Indirect Effect | Total Effect |
| M01: Perceived patient safety | −0.10 | NA | −0.10 | −2.41* | NA | **−2.41*** |
| P01: Transformational leadership | −0.66* | −0.02 | **−0.69*** | 1.03* | −1.21* | −0.19 |
| P02: Job satisfaction | −0.06 | 0.04 | −0.02 | 0.59* | 0.27* | **0.85*** |
| P03: Patient-related burnout | 0.20* | 0.01 | **0.21*** | 1.06* | −1.02* | 0.04 |
| P04: Individual influence on patient safety [a] | −0.20* | −0.02 | **−0.22*** | 1.89* | −1.54* | **0.35*** |
| P05: Staffing | −0.56* | −0.01 | **−0.57*** | 0.23* | −0.62* | **−0.39*** |
| P06: Organizational learning | 0.45* | −0.04 | **0.41*** | −0.67* | 1.18* | **0.51*** |
| P07: Feedback and communication about error | 0.49* | −0.02 | **0.47*** | −0.44* | −0.50* | **−0.94*** |
| P08: Handoffs and transitions | −0.56* | −0.04 | **−0.59*** | 1.24* | −1.08* | **0.15*** |
| P09: Hospital management support for patient safety | 0.42* | −0.03 | **0.39*** | 0.10 | −1.16* | **−1.06*** |
| P10: Hospital management behaviours related to patient safety | −0.06 | 0.03 | −0.03 | −0.91* | 0.77* | −0.14 |
| P11: Direct supervisor support for patient safety | 0.66* | −0.02 | **0.63*** | −0.43* | 0.60* | 0.17 |
| P12: Direct supervisor behaviours related to patient safety | −1.05* | 0.02 | **−1.04*** | 1.37* | −0.75* | **0.62*** |

Note: All effects are standardized. *$p < 0.05$, two-tailed significance based on bootstrapping (500 samples); NA – not applicable. a – Single item. Statistically significant total effects are marked with bold. Due to rounding, the sum of direct and indirect effects may not exactly match the total effects.

significant effect on Documentation Quality (standardized direct effect −2.40, p < 0.05). Correspondingly, all the predictors (P01-P12) had significant indirect effects on Documentation Quality. In case of physicians, the effect of Perceived Patient Safety (M01) on Documentation Quality was not statistically significant (−0.10, p > 0.05) and consequently none of the predictors (P01-P12) had significant indirect effect on Documentation Quality.

For both professional groups, the model explained >90% of the variance in the mediator, Perceived Patient Safety (M1), and >75% of the variance in the latent factor Documentation Quality (Table 4). For the six specific aspects of Documentation Quality, the largest proportion of variance explained was for O1-O3, associated with the quality of medication prescriptions. The model explained up to 11% of the variance related to the quality of the medication administration chart (O4-O6).

## Discussion

In this study, we aimed to evaluate organizational factors that may predict Documentation Quality in hospital settings. We developed theoretical model and addressed the research questions in a complex study, consisting of two data sources – a survey of frontline clinical personnel regarding leadership, job satisfaction, patient-related burnout, and patient safety culture; and data generated by retrospective chart review of patient records in participating departments in the same time period. Consequently, we were able to combine the survey data, aggregated at the department level, with the chart review data in the path model to answer the research questions.

### The documentation quality is associated with leadership, job satisfaction, patient-related burnout and patient safety culture (Q1)

Most of the predictors included in the analysis, organizational factors based on survey of physicians and nurses, had significant effect on Documentation Quality. The model explained a sufficient proportion of variance in medication Documentation Quality using the survey data from frontline workers. Undoubtedly, there are other factors in play, which were not included in this analysis. A recent meta-analysis evaluated the association between transformational leadership and the quality of patient care, as perceived by nurses [33]. While the meta-analysis found a positive total effect, individual studies reported varying direct and indirect effects, indicating the presence of confounding and mediating factors (organizational, cultural, individual). The model explained more variance in the quality of medication prescription charts (O1-O3), compared to medication administration charts (O4-O5). Prescription charts, mainly filled out by physicians, and perhaps that's why, the survey data from physicians explained more variance in medication prescription charts, than survey data from nurses. Other significantly factors likely affect the quality of administration charts [34,35].

Our analysis demonstrated significant effects of various organizational factors on Documentation Quality, and consequently, the answer to the first research question seems positive. However, these effects are clearly more complex

**Table 4. Proportion of variance in mediator and outcome variables explained by the model.**

|  | Physicians (R²) | Nurses (R²) |
|---|---|---|
| M1-Perceived patient safety | 0.93 | 0.95 |
| Documentation Quality (latent factor) | 0.79 | 0.76 |
| O1. Date, Time, Signature | 0.71 | 0.58 |
| O2. Prescription Completeness | 0.14 | 0.16 |
| O3. Legibility of prescription chart | 0.16 | 0.17 |
| O4. First medication administration chart | 0.05 | 0.09 |
| O5. Transcription of adjustments | 0.08 | 0.11 |
| O6. Legibility of administration chart | 0.05 | 0.08 |

than our model could test, with unexpected effect directions, inconsistencies between professional groups, and contradicting direct and indirect effects. These findings underscore the need for further studies to explore these complex relationships.

### Perceived patient safety may play a mediating role (Q2)

Perceived Patient Safety had a significant effect on Documentation Quality in the analysis involving nurses, but this effect was not significant for physicians. Consequently, all predictors had significant indirect effects on Documentation Quality for nurses, indicating that outcome measure Perceived Patient Safety may play a mediating role in this group, but not for physicians. The negative mediation effect observed for nurses is an unexpected finding that requires further exploration. Because the effect was negative, most independent variables exhibited contradicting direct and indirect effects, cancelling each other out, resulting in lower or non-significant total effects. For instance, Transformational Leadership had a strong positive direct effect (1.03, $p < 0.05$) and a negative indirect effect mediated through Perceived Patient Safety ($-1.21$, $p < 0.05$), resulting in a non-significant total effect ($-0.19$, $p = 0.12$). Similarly, a meta-analysis on nurses' perceptions of transformational leadership and quality of patient care found varying direct and mediated effects [33]. One possible explanation may be overconfidence in environments with highly perceived safety, where strong team support allows members to catch and correct possible mistakes. Employees may feel less urgency to meticulously document medications, prioritizing direct patient care. This may be particularly pronounced if high perceived safety coincides with increased workload, further limiting the time available for documentation. Clearly, the relationship between Documentation Quality and perceived patient safety is complex and requires further study. A qualitative study exploring nurses' perceptions also revealed a complex interplay of a wide range of factors involved in medication errors [34].

### There are considerable differences in survey results of physicians and nurses (Q3)

As expected, and well-supported by the available literature, our result demonstrated significant differences between professional groups [25]. The direct effects observed in two professional groups were largely contradicting. For example, while for nurses the effects of Job Satisfaction (P02), Individual Influence on Patient Safety (P04) and Direct Supervisor Behaviours Related to Patient Safety (P12) were positive, the same factors had negative or non-significant (P02) effects for physicians. Furthermore, in case of physicians, Perceived Patient Safety did not have a significant mediating effect on Documentation Quality, the one we observed with nurses. These findings likely reflect broader differences in professional identity and roles, as well as differences in perceptions consistently demonstrated in studies from various countries [24,25,36]. Such differences between physicians and nurses, potentially linked to distinct professional responsibilities, management structures, and cultures, should be considered when planning research projects, tailored interventions, as well as when interpreting or comparing results across professional groups.

### Practical implications

This study highlights the role of organizational factors – such as leadership, job satisfaction, patient-related burnout, and patient safety culture – for the medication documentation quality. These findings suggest that efforts to improve documentation quality should extend beyond narrow technical solutions and consider organizational and behavioural interventions. Strengthening leadership engagement, fostering a positive safety culture, and addressing staff well-being could contribute to more reliable medication documentation practices. Moreover, our results indicate that professional groups may respond differently to these factors. This suggests that tailored interventions, accounting for professional differences, may be more effective than uniform, one-size-fits-all approaches.

By considering these findings, healthcare institutions can address documentation errors, improve information flow, and ultimately enhance patient safety. Future research should evaluate the effectiveness of such interventions and explore additional factors influencing documentation quality in clinical settings.

## Limitations

One limitation of the study is the level of analysis. The data from frontline staff surveys were aggregated at the department level to link it with the chart review data, which is on the patient level. Although this aggregation method is frequently used in health services research, it may mask individual and intra-departmental variations. Additionally, we were unable to consider the clinical area of participating departments, which may have introduced additional variance either in the survey or chart review data. Moreover, the interpretation of the study findings should be made with the response rate in mind. While our sample size allowed for robust analyses, the potential for response bias cannot be excluded, which could affect the generalizability of our findings. While the results support the theoretical model, which demonstrated acceptable fit with the data, a different theoretical model can be developed and tested based on other theoretical considerations. As the variance in Documentation Quality was not fully explained by our model, there must be other influential factors at play. Further studies could include factors related to the clinical area, organization or even the healthcare (e.g., workload, case mix, team or cultural dynamics), as well as individual factors related to frontline professionals (e.g., years of experience) and characteristics of patients (e.g., socioeconomic status or disease severity). Finally, the cross-sectional design of our study limits our ability to establish causal relationships between the explored factors and the quality of medical documentation. Longitudinal studies would be valuable to provide deeper insights into how these factors affect documentation quality over time.

## Conclusions

This study demonstrated that certain aspects of medication documentation quality are associated with organizational factors, such as leadership, job satisfaction, patient-related burnout and patient safety culture. Still, there is a large portion of unexplained variance, which may be influenced by other organizational factors, as well as factors associated with clinical area, clinical professionals, or patients. Our analyses revealed significant differences between nurses' and physicians' perceptions of leadership, job satisfaction, patient-related burnout and safety culture. These differences were well reflected in possibility to explain Documentation Quality. In case of nurses, Perceived Patient Safety had a significant mediating effect on Documentation Quality, which was not the case for physicians. Finally, we found that the organizational factors included in our analyses predicted the quality of prescription charts considerably better than the quality of administration charts.

## Supporting information

**S1 Appendix. S1 Figure.** The model used in the path analysis.
(PDF)

**S2 Appendix. S2 Table.** The fit of the unconstrained multi-group model used in the path analysis.
(PDF)

## Acknowledgments

We thank the members of the advisory board for their valuable advice at various stages of the project: Prof. Johannes Giehl (Competence Centre Quality Assurance/Management (KTQ), Medical Service of Statutory Healthcare Assurance in Germany), Prof. Ulrich Jaehde (Institute of Pharmacy, University of Bonn), Dr. Constanze Lessing (Berlin), Dr. Barbara Strohbuecker (Deutscher Pflegerat, Cologne), Prof. David Schwappach (Swiss Patient Safety Foundation, Zurich), Prof. Petra Thürmann (University Witten/Herdecke, Chair of Clinical Pharmacology; HELIOS University Clinic Wuppertal). We also thank Dr. Martina Michaelis (Institute of Occupational and Social Medicine and Health Services Research Tuebingen and Research Centre for Occupational and Social Medicine (FFAS), Freiburg) for the valuable support in developing the survey. Last but not least, we would like to thank all study participants. We acknowledge the support of the hospital

management and workers´ representatives in both hospitals, the efforts of study coordinators in participating departments and units to facilitate data collection, and the respondents for their effort and time to fill in the surveys.

**WorkSafeMed Project Consortium**

During the study members of the WorkSafeMed Project Consortium were: Luntz E, Rieger MA (Project lead), Sturm H, Wagner A (Institute of Occupational and Social Medicine and Health Services Research, University Hospital of Tuebingen), Hammer A, Manser T (Institute for Patient Safety, University Hospital Bonn), Martus P (Institute for Clinical Epidemiology and Applied Biometry, University Hospital of Tuebingen), Holderried M (University Hospital Tuebingen).

## Author contributions

**Conceptualization:** Anke Wagner, Tanja Manser, Monika A. Rieger, Peter Martus, Antje Hammer.

**Data curation:** Nikoloz Gambashidze, Peter Martus, Antje Hammer.

**Formal analysis:** Nikoloz Gambashidze.

**Investigation:** Anke Wagner, Tanja Manser, Monika A. Rieger, Antje Hammer.

**Methodology:** Nikoloz Gambashidze, Anke Wagner, Tanja Manser, Monika A. Rieger, Peter Martus, Antje Hammer.

**Project administration:** Tanja Manser, Monika A. Rieger.

**Supervision:** Anke Wagner, Tanja Manser, Monika A. Rieger, Antje Hammer.

**Visualization:** Nikoloz Gambashidze.

**Writing – original draft:** Nikoloz Gambashidze, Anke Wagner, Antje Hammer.

**Writing – review & editing:** Nikoloz Gambashidze, Anke Wagner, Tanja Manser, Monika A. Rieger, Peter Martus, Antje Hammer.

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
