## [Decision Letter · Decision Letter 0]

29 Dec 2024

Dear Dr. Gambashidze,

Thank you for submitting your manuscript to PLOS ONE. After careful consideration, we feel that it has merit but does not fully meet PLOS ONE’s publication criteria as it currently stands. Therefore, we invite you to submit a revised version of the manuscript that addresses the points raised during the review process.

**ACADEMIC EDITOR:**

The reviewers' comments below are critical and should be addressed carefully.

We look forward to receiving your revised manuscript.

Kind regards,

Hossam Elamir, MSc

Academic Editor

PLOS ONE

Journal Requirements:

The “WorkSafeMed“ study was funded by the Federal Ministry of Education and Research (FKZ 01GY1325A and 01GY1325B). In addition, the work of the Institute of Occupational and Social Medicine and Health Services Research Tuebingen is supported by an unrestricted grant of the Employers´ Association of the Metal and Electric Industry Baden-Wuerttemberg (Suedwestmetall). We acknowledge additional financial support by the German Research Foundation and the Open Access Publishing Fund of the University of Tuebingen, and the administrative support by the DLR Project Management Agency. 

4. In the online submission form, you indicated that due to data security and privacy regulations, data from the WorkSafeMed study cannot be shared publicly. However, data will be securely stored in compliance with national and regional data protection standards. Researchers who meet the criteria for access to confidential data may request access from AH, subject to reasonable conditions.

5. We note that you have indicated that there are restrictions to data sharing for this study. For studies involving human research participant data or other sensitive data, we encourage authors to share de-identified or anonymized data. However, when data cannot be publicly shared for ethical reasons, we allow authors to make their data sets available upon request. For information on unacceptable data access restrictions, please see http://journals.plos.org/plosone/s/data-availability#loc-unacceptable-data-access-restrictions. 

6. Your abstract cannot contain citations. Please only include citations in the body text of the manuscript, and ensure that they remain in ascending numerical order on first mention.

Reviewers' comments:

Reviewer's Responses to Questions

**Comments to the Author**

1. Is the manuscript technically sound, and do the data support the conclusions?

Reviewer #1: Yes

Reviewer #2: Yes

Reviewer #3: Yes

2. Has the statistical analysis been performed appropriately and rigorously?

Reviewer #1: Yes

Reviewer #2: Yes

Reviewer #3: Yes

3. Have the authors made all data underlying the findings in their manuscript fully available?

Reviewer #1: Yes

Reviewer #2: Yes

Reviewer #3: Yes

4. Is the manuscript presented in an intelligible fashion and written in standard English?

Reviewer #1: Yes

Reviewer #2: Yes

Reviewer #3: Yes

Reviewer #1: Thank you for submitting your study to our journal.

Your study addresses a crucial issue in healthcare—medication documentation quality and its organizational predictors—which is an essential contribution to patient safety literature. The methodology is robust, using validated tools and multi-group path analysis, and the dataset is comprehensive and valuable.

To further enhance your manuscript, I have some suggestions for improvement:

- There are minor grammatical errors, such as "To what extend," which should be "extent." A thorough review of the text is recommended to address these issues.

- While the sample size is sufficient, the 39.6% response rate may introduce response bias. It would be helpful to discuss how this might affect the generalizability of your findings.

- Aggregating survey data at the department level could dilute individual-level variations. Please provide a clear justification for why this approach was chosen.

- The negative perceived patient safety 'mediation effect' observed for nurses may require further exploration.

- The positive job satisfaction for nurses but not physicians highlights professional differences, which may require more clarification and may help in tailored intervention.

- While the theoretical model is sound, it needs to fully explain the variance in outcomes. Consider refining the model by including additional predictors like workload, team dynamics, or individual factors like years of experience.

- While the ethics statement is thorough, the decision to restrict data availability does not align with PLOS ONE's data-sharing policy. Consider providing anonymized data or offering a more precise justification for the restrictions.

Thank You.

Reviewer #2: The study, titled "A multi-group path analysis of medication documentation quality using cross-sectional survey data: Impact of leadership, job satisfaction, patient-related burnout, and patient safety culture," examines a crucial area of healthcare: the quality of medication documentation and its relationship with organizational and individual factors. The research provides a significant contribution to the understanding of how leadership, job satisfaction, burnout, and safety culture influence documentation practices, which are directly linked to patient safety and the prevention of adverse medication events.

The methodological framework is robust, employing a multi-group path analysis to explore the interplay between organizational factors and documentation quality. By using data collected from 24 departments across two German university hospitals, the study incorporates survey responses from frontline workers—282 physicians and 417 nurses—and retrospective reviews of 802 patient records. This dual approach allows for a comprehensive analysis that links staff perceptions to objective outcomes in documentation quality. Established instruments, such as the Transformational Leadership Inventory and the Copenhagen Psychosocial and Burnout Inventories, lend credibility to the survey data, while the MediDoq instrument ensures a standardized and detailed assessment of medication documentation.

One of the study's strengths is its focus on comparing the effects of the examined factors across professional groups. The findings reveal significant differences between physicians and nurses, particularly in how perceived patient safety mediates the relationship between organizational factors and documentation quality. For nurses, perceived patient safety played a substantial mediating role, while for physicians, it did not. These nuanced findings highlight the importance of tailoring interventions to the specific needs and dynamics of professional groups within healthcare settings.

Despite its strengths, the study has some limitations. The high level of aggregation of survey data at the department level may obscure individual-level variations and introduce potential biases. Additionally, the high response rate for surveys (approximately 39.6%) is commendable, but the lack of long-term follow-up limits insights into sustained impacts. Furthermore, while the multi-group path analysis demonstrates a good fit with the data, the study acknowledges that other unexplored factors may influence documentation quality, leaving room for future research to build on these findings.

In conclusion, this study provides a valuable framework for understanding the complex relationships between organizational factors and medication documentation quality. Its methodological rigor and practical implications make it a strong candidate for publication. With its findings, the study not only advances academic understanding of patient safety culture but also offers actionable insights for improving healthcare practices. Nonetheless, addressing the stated limitations and expanding the scope of analysis in future research would further enhance the robustness and applicability of the findings.

Reviewer #3: GENERAL COMMENTS

o The paper is well-written and encounters a very important topic. Some editing points need to be checked about grammar and punctuation.

DETAILED COMMENTS:

INTRODUCTION: It covers the background literature and study aims and objectives. However, it would be informative to document any previous similar study predicting individual or organizational factors affecting the quality of medication documentation. In addition, it is important to highlight at the end of the introduction why you are conducting the current study (e.g. lack of literature, what this study will add or fulfill the gaps in the literature).

o Page 3, line 46: "A recent metanalysis found, that for hospital settings, as high as 19% of all hospitalized patients…." When checking the reference used (Ref 1), the paper was published in 2017 which is not recent. I suggest you either update the ref with a more recent meta-analysis or paraphrase the sentence. Please remove the comma after "found."

o Page 3, line 69: Please check the grammar; the word "effects" should be changed to the verb "affects".

o Page 4, lines 78-79: Please move ref (10) to the end of the sentence.

o Page 4, lines 86-87: Please change " The model, developed based on………, is presented on figure 1" to the model, developed based on………, is presented in figure 1"

METHODS

o In the methods section, please clarify the following:

1. How the conceptual framework was developed. Were there any validation steps undertaken for the framework?

2. For the MediDocQ Instrument, not all the original criteria described in ref 7 were used in this study. So how this will affect the analysis and interpretation of results?

3. Did you carry out any sample size calculations?

4. Please clarify the sampling sampling strategy. The HCPs were selected by convenience as mentioned in the manuscript, however, how were the departments selected?

o Please write the inclusion/exclusion criteria in the methods section.

RESULTS:

Well presented. However, one point is about table 1 (page 9): the results from HSPSC were expressed as mean (+/-SD), although the AHRQ's guide for survey analysis is to express the results by positive response rate. How would this affect the interpretation of the results?

DISCUSSION

o Page 14, line 314: the net difference of 1.03 and -1.21 is -0.18 not -0.19. please correct.

o Page 14, line 320: Please insert ref after the first sentence.

o Page 14, line 331: Please change the sentence "…..should be considered when planning research or improvement projects,…" to sentence "…..should be considered when planning research or improving projects,…"

o What are the implications of this study on the clinical practice? Please indicate in the manuscript.

**Do you want your identity to be public for this peer review?** For information about this choice, including consent withdrawal, please see our Privacy Policy

Reviewer #1: **Yes: ** Dr. Ahmed Newera

Reviewer #2: **Yes: ** Izabella Uchmanowicz

Reviewer #3: No

---

## [Author Response · Author response to Decision Letter 1]

26 Mar 2025

Dear Editors and Reviewers,

We sincerely appreciate the time and effort you have taken to evaluate our manuscript. We are grateful for the constructive feedback, which has helped us refine and improve our work. Below, we provide a detailed, point-by-point response to all comments.

We hope that our revisions have adequately addressed all concerns and that the improved manuscript meets the expectations of the journal. We thank you again for your thoughtful review and look forward to your feedback.

Comments from the editorial office

Response: Thank you for highlighting the importance of adhering to PLOS ONE's style requirements. We have thoroughly reviewed the guidelines provided by the journal and have adjusted our manuscript and file names accordingly to ensure full compliance. We hope you will find the revised documents meeting the set requirements.

The „WorkSafeMed“ study was funded by the Federal Ministry of Education and Research (FKZ 01GY1325A and 01GY1325B). In addition, the work of the Institute of Occupational and Social Medicine and Health Services Research Tuebingen is supported by an unrestricted grant of the Employers´ Association of the Metal and Electric Industry Baden-Wuerttemberg (Suedwestmetall). We acknowledge additional financial support by the German Research Foundation and the Open Access Publishing Fund of the University of Tuebingen, and the administrative support by the DLR Project Management Agency.

Response: We confirm that the funders had no role in the study design, data collection and analysis, decision to publish, or preparation of the manuscript. This statement will be included in our cover letter, as requested.

Response: We have revised the manuscript and confirm that the ethics statement now exclusively appears in the Methods section.

4. In the online submission form, you indicated that due to data security and privacy regulations, data from the WorkSafeMed study cannot be shared publicly. However, data will be securely stored in compliance with national and regional data protection standards. Researchers who meet the criteria for access to confidential data may request access from AH, subject to reasonable conditions.

Response: We understand and support the journal's commitment to ensuring that data underlying the findings are accessible to enhance transparency and reproducibility. In the case of the WorkSafeMed study, the data involves sensitive personal information that is strictly governed by national and regional data protection laws. This dataset encompasses survey responses from clinical personnel and the reviews of the records of individual patients. Public deposition of this data would violate the ethical approval provided by our research ethics board, which mandates stringent measures to protect participant privacy and ensure compliance with legal requirements. A copy and English translation of the ethical approval is among the submission documents.

Consequently, while we cannot deposit the data in a public repository or include it directly within the manuscript or as supplementary information, we are committed to making the data available to researchers who meet stringent criteria for access to confidential data. This process will be managed through a formal data access request to the Institute for Patient Safety, Bonn (ifps@ukbonn.de), which will ensure any data sharing complies with the necessary confidentiality and ethical standards.

We have revised the Manuscript (declarations, availability of data and materials) and provided our rationale for this exemption and the mechanisms for data access. We kindly request your consideration and approval in accordance with the journal's policy for such exceptions.

5. We note that you have indicated that there are restrictions to data sharing for this study. For studies involving human research participant data or other sensitive data, we encourage authors to share de-identified or anonymized data. However, when data cannot be publicly shared for ethical reasons, we allow authors to make their data sets available upon request. For information on unacceptable data access restrictions, please see http://journals.plos.org/plosone/s/data-availability#loc-unacceptable-data-access-restrictions.

Response: Please see the response to the previous comment for detailed discussion on data availability. The constraints imposed by the ethics committee are due to the sensitive nature of the data, prevent us from publicly sharing the data. We have updated the "Availability of Data and Materials" section of our manuscript to reflect these restrictions, indicating process for accessing the data upon reasonable request.

6. Your abstract cannot contain citations. Please only include citations in the body text of the manuscript, and ensure that they remain in ascending numerical order on first mention.

Response: We have reviewed the abstract, ensuring that it does not include any citations. We have also reviewed the entire manuscript to confirm that citations in the body text are in ascending numerical order on first mention.

Reviewers' comments:

Reviewer's Responses to Questions

Comments to the Author

1. Is the manuscript technically sound, and do the data support the conclusions?

Reviewer #1: Yes

Reviewer #2: Yes

Reviewer #3: Yes

2. Has the statistical analysis been performed appropriately and rigorously?

Reviewer #1: Yes

Reviewer #2: Yes

Reviewer #3: Yes

3. Have the authors made all data underlying the findings in their manuscript fully available?

Reviewer #1: Yes

Reviewer #2: Yes

Reviewer #3: Yes

4. Is the manuscript presented in an intelligible fashion and written in standard English?

Reviewer #1: Yes

Reviewer #2: Yes

Reviewer #3: Yes

5. Review Comments to the Author

Reviewer #1:

Thank you for submitting your study to our journal.

Your study addresses a crucial issue in healthcare—medication documentation quality and its organizational predictors—which is an essential contribution to patient safety literature. The methodology is robust, using validated tools and multi-group path analysis, and the dataset is comprehensive and valuable. To further enhance your manuscript, I have some suggestions for improvement.

Response: We appreciate your encouraging evaluation, as well as careful recommendations to improve the manuscript. Below we address each of them individually.

5.1.1. There are minor grammatical errors, such as "To what extend," which should be "extent." A thorough review of the text is recommended to address these issues.

Response: Thank you for pointing out the minor grammatical issues present in our manuscript. We have conducted a thorough review of the text to correct these issues, including the specific error you noted. We hope you will find that the revisions have improved the overall clarity and readability of our manuscript.

5.1.2. While the sample size is sufficient, the 39.6% response rate may introduce response bias. It would be helpful to discuss how this might affect the generalizability of your findings.

Response: We agree and acknowledge that while our sample size is sufficient for analysis, the response bias can not be excluded, which might limit the generalizability of our findings. In response, we have extended the discussion section to address this issue explicitly.

5.1.3. Aggregating survey data at the department level could dilute individual-level variations. Please provide a clear justification for why this approach was chosen.

Response: Thank you for raising the issue of the data aggregation. In our study, we utilized two primary data sources: surveys, conducted at the individual employee level, and record reviews, at the individual patient level. Due to the nature of these data sets, the most viable option to match and analyze these simultaneously was to aggregate the data at the department level. This approach is justified as the constructs evaluated in the employee survey, such as safety culture or job satisfaction, are considered to be group-level constructs, typically analyzed at the department or even hospital level in health services research.

We recognize the importance of clarifying this methodological decision and have revised the methods section to make this rationale more transparent.

5.1.4. The negative perceived patient safety 'mediation effect' observed for nurses may require further exploration.

Response: We appreciate you highlighting this interesting and unexpected mediation effect. The negative relationship between Perceived Patient Safety and Documentation Quality for nurses remains an open question that requires further investigation. While we have provided some possible explanations—such as overconfidence in highly safe environments or the impact of workload limiting documentation time—the underlying mechanisms are not yet fully understood. It is also possible that additional factors not captured in our study contribute to this effect. We have expanded the Discussion section to acknowledge these possibilities and emphasize the need for further research to explore this complex relationship.

5.1.5. The positive job satisfaction for nurses but not physicians highlights professional differences, which may require more clarification and may help in tailored intervention.

Response: Thank you for highlighting the significant effect of job satisfaction for nurses but not physicians. That this point is clearly underlined in the revised discussion.

5.1.6. While the theoretical model is sound, it needs to fully explain the variance in outcomes. Consider refining the model by including additional predictors like workload, team dynamics, or individual factors like years of experience.

Response: We appreciate your suggestion to enhance our theoretical model by including additional predictors. While we agree that this could enrich our findings, we are constrained by the original study design and the data currently available to us. We have extended the limitation section with a recommendation for future studies to include additional factors, which may extend our understanding of the complex interplay between organizational factors and the quality of medical documentation.

5.1.7. While the ethics statement is thorough, the decision to restrict data availability does not align with PLOS ONE's data-sharing policy. Consider providing anonymized data or offering a more precise justification for the restrictions.

Response: Thank you for your feedback on data availability. As noted in our response to the Editor's Comment #4, strict constraints from the ethical approval prevent sharing data publicly. We have detailed these constrains and the procedure for obtaining the data in the Data Availability statement accordingly.

Reviewer #2:

The study, titled "A multi-group path analysis of medication documentation quality using cross-sectional survey data: Impact of leadership, job satisfaction, patient-related burnout, and patient safety culture," examines a crucial area of healthcare: the quality of medication documentation and its relationship with organizational and individual factors. The research provides a significant contribution to the understanding of how leadership, job satisfaction, burnout, and safety culture influence documentation practices, which are directly linked to patient safety and the prevention of adverse medication events.

The methodological framework is robust, employing a multi-group path analysis to explore the interplay between organizational factors and documentation quality. By using data collected from 24 departments across two German university hospitals, the study incorporates survey responses from frontline workers—282 physicians

---

## [Decision Letter · Decision Letter 1]

9 Jun 2025

Dear Dr. Gambashidze,

Thank you for submitting your manuscript to PLOS ONE. After careful consideration, we feel that it has merit but does not fully meet PLOS ONE’s publication criteria as it currently stands. Therefore, we invite you to submit a revised version of the manuscript that addresses the points raised during the review process.

We look forward to receiving your revised manuscript.

Kind regards,

Hossam Elamir, MSc

Academic Editor

PLOS ONE

**Journal Requirements:**

Reviewers' comments:

Reviewer's Responses to Questions

**Comments to the Author**

Reviewer #1: All comments have been addressed

Reviewer #4: (No Response)

2. Is the manuscript technically sound, and do the data support the conclusions?

Reviewer #1: Yes

Reviewer #4: Yes

3. Has the statistical analysis been performed appropriately and rigorously?

Reviewer #1: Yes

Reviewer #4: Yes

4. Have the authors made all data underlying the findings in their manuscript fully available?

Reviewer #1: Yes

Reviewer #4: Yes

5. Is the manuscript presented in an intelligible fashion and written in standard English?

Reviewer #1: Yes

Reviewer #4: Yes

**Reviewer #1: ** The authors have thoroughly addressed all reviewer and editorial comments, demonstrating rigorous revisions, including methodological clarifications, enhanced discussion of limitations/practical implications, and strict adherence to ethical and formatting guidelines.

This exemplary manuscript contributes to understanding organizational factors in medication documentation quality, with robust methodology and clear translational impact for healthcare improvement.

The authors' responsiveness to feedback underscores their commitment to scholarly excellence.

Best regards

**Reviewer #4:**  This study offers valuable insights of an aspect that overlooked of patient safety. The use of validated instruments and multi-group path analysis is a notable strength of this study.

The former reviewers provided valuable and well-considered comments. The authors have responded with clear justifications and making appropriate revisions in nearly all cases.

Here are some few comments:

Project: brief explanation of the project. “Working Conditions, Safety Culture and Patient Safety in Hospitals the Safety of the Medication Process (WorkSafeMed) (would be valuable for the readers, if it is added to method section

In the study design section, the term “an appropriate sample size” is used. More clarification is needed about how the author recruited the participants tell they reached 995 participants and which sample size calculations was used?

Although this comment was addressed by another reviewer, but still the clarification by the author in the edited manuscript is not clear.

Figure 1

Ensure Figure 1 (conceptual framework) is clearly legible and includes all variable labels used in the manuscript text. Also, consider including a short narrative walkthrough of the model for readers less familiar with path analysis.

Terminology Consistency

At times, the manuscript alternates between terms like “documentation quality,” “documentation practices,” and “medication documentation.” Ensure consistency

**Do you want your identity to be public for this peer review?** For information about this choice, including consent withdrawal, please see our Privacy Policy

Reviewer #1: **Yes: ** Ahmed Newera

Reviewer #4: No

---

## [Author Response · Author response to Decision Letter 2]

24 Jul 2025

Dear Editors and Reviewers,

We sincerely thank you for your continued engagement with our manuscript. We are grateful for the thoughtful and constructive comments provided in the second round of review. These helped us to further improve the clarity and consistency of the manuscript.

Below, we provide a point-by-point response outlining the revisions we have made. We hope that the updated manuscript now meets requirements for publication in PLOS ONE. Thank you once again for your time and consideration.

Comments from the editorial office

Journal Requirements:

Response: We have reviewed all references cited in the manuscript to ensure that none have been retracted. No retracted articles are included in the current version of the manuscript. Therefore, no changes to the reference list were necessary in this regard.

Reviewers' comments:

Reviewer's Responses to Questions

Comments to the Author

1. If the authors have adequately addressed your comments raised in a previous round of review and you feel that this manuscript is now acceptable for publication, you may indicate that here to bypass the “Comments to the Author” section, enter your conflict of interest statement in the “Confidential to Editor” section, and submit your "Accept" recommendation.

Reviewer #1: All comments have been addressed

Reviewer #4: (No Response)

2. Is the manuscript technically sound, and do the data support the conclusions?

Reviewer #1: Yes

Reviewer #4: Yes

3. Has the statistical analysis been performed appropriately and rigorously?

Reviewer #1: Yes

Reviewer #4: Yes

4. Have the authors made all data underlying the findings in their manuscript fully available?

Reviewer #1: Yes

Reviewer #4: Yes

5. Is the manuscript presented in an intelligible fashion and written in standard English?

Reviewer #1: Yes

Reviewer #4: Yes

6. Review Comments to the Author

Reviewer #1

The authors have thoroughly addressed all reviewer and editorial comments, demonstrating rigorous revisions, including methodological clarifications, enhanced discussion of limitations/practical implications, and strict adherence to ethical and formatting guidelines.

This exemplary manuscript contributes to understanding organizational factors in medication documentation quality, with robust methodology and clear translational impact for healthcare improvement.

The authors' responsiveness to feedback underscores their commitment to scholarly excellence.

Response: We sincerely thank the reviewer for the positive and encouraging feedback. We appreciate your recognition of our efforts to thoroughly revise the manuscript and address all comments. Thank you for your support throughout the review process.

Reviewer #4:

This study offers valuable insights of an aspect that overlooked of patient safety. The use of validated instruments and multi-group path analysis is a notable strength of this study.

The former reviewers provided valuable and well-considered comments. The authors have responded with clear justifications and making appropriate revisions in nearly all cases.

Response: We thank Reviewer #4 for the thoughtful evaluation and kind remarks. We appreciate the recognition of our methodological approach and the revisions made in response to prior comments.

#4.1. Project: brief explanation of the project. “Working Conditions, Safety Culture and Patient Safety in Hospitals the Safety of the Medication Process (WorkSafeMed) (would be valuable for the readers, if it is added to method section

Response: Thank you for the opportunity to clarify the scope of the project. We have revised the Methods section to include the following sentence: “The project aimed to examine the influence of working conditions and safety culture on patient safety, with a focus on medication process in hospitals.”

#4.2. In the study design section, the term “an appropriate sample size” is used. More clarification is needed about how the author recruited the participants tell they reached 995 participants and which sample size calculations was used?

Although this comment was addressed by another reviewer, but still the clarification by the author in the edited manuscript is not clear.

Response: Thank you for your comment. As noted in our previous revision, we did not perform a formal a priori sample size calculation. Instead, we aimed for comprehensive recruitment of clinical staff from all eligible departments to ensure sufficient sample size for multivariate analysis. We have now clarified this point in the Methods section: “We invited all physicians and nurses from the eligible departments to participate in the survey. An a priori sample size calculation was not conducted.”

#4.3. Ensure Figure 1 (conceptual framework) is clearly legible and includes all variable labels used in the manuscript text. Also, consider including a short narrative walkthrough of the model for readers less familiar with path analysis.

Response: Thank you for this very helpful observation. We carefully reviewed the manuscript and the figure to ensure that all variable labels used in the path model are consistent with the terminology in the main text. We have revised Figure 1 accordingly. To support interpretation, we also added the following brief explanatory passage to the Methods section: “In the model, organizational factors such as transformational leadership, job satisfaction, patient-related burnout, and various aspects of patient safety culture were included as predictors of medication documentation quality. Perceived patient safety was specified as a potential mediator of the relationship between organizational predictors and documentation quality. Separate models were estimated for nurses and physicians using multi-group path analysis to explore profession-specific relationships.”

#4.4. Terminology Consistency: At times, the manuscript alternates between terms like “documentation quality,” “documentation practices,” and “medication documentation.” Ensure consistency

Response: Thank you for pointing this out. We reviewed the manuscript for consistency in terminology and revised instances where alternate terms were used. We now consistently use “(medication) documentation quality” throughout the manuscript to maintain clarity and coherence.

7. PLOS authors have the option to publish the peer review history of their article (what does this mean?). If published, this will include your full peer review and any attached files.

Do you want your identity to be public for this peer review? For information about this choice, including consent withdrawal, please see our Privacy Policy.

Reviewer #1: Yes: Ahmed Newera

Reviewer #4: No

---

## [Editor Report · Decision Letter 2]

3 Aug 2025

A multi-group path analysis of medication documentation quality using cross-sectional survey data: Impact of leadership, job satisfaction, patient-related burnout, and patient safety culture

PONE-D-24-43570R2

Dear Dr. Gambashidze,

We’re pleased to inform you that your manuscript has been judged scientifically suitable for publication and will be formally accepted for publication once it meets all outstanding technical requirements.

Kind regards,

Hossam Elamir, MSc

Academic Editor

PLOS ONE
---

## [Editor Report · Acceptance letter]

PONE-D-24-43570R2

PLOS ONE

Dear Dr. Gambashidze,

I'm pleased to inform you that your manuscript has been deemed suitable for publication in PLOS ONE. Congratulations! Your manuscript is now being handed over to our production team.

Kind regards,

on behalf of

Dr. Hossam Elamir

Academic Editor

PLOS ONE